# Assessment of Resilience of the Hellenic Navy Seals by Electrodermal Activity during Cognitive Tasks

**DOI:** 10.3390/ijerph18084384

**Published:** 2021-04-20

**Authors:** Stamatis Mourtakos, Georgia Vassiliou, Konstantinos Kontoangelos, Christos Papageorgiou, Anastasios Philippou, Fragkiskos Bersimis, Nikolaos Geladas, Michael Koutsilieris, Labros S. Sidossis, Charalampos Tsirmpas, Charalabos Papageorgiou, Konstantina G. Yiannopoulou

**Affiliations:** 1Department of Psychiatry, University of Athens School of Medicine, 11528 Athens, Greece; stmourt@gmail.com (S.M.); g.vassiliou02@gmail.com (G.V.); kontoangel@med.uoa.gr (K.K.); chpapag@med.uoa.gr (C.P.); 2Medical School, Department of Physiology, National and Kapodistrian University of Athens, 11527 Athens, Greece; tfilipou@med.uoa.gr (A.P.); mkoutsil@med.uoa.gr (M.K.); 3251 Air Force General Hospital, 11527 Athens, Greece; chrispapageorgio@gmail.com; 4Department of Supply Chain Management, Agricultural University of Athens, 11855 Athens, Greece; fbersim3@gmail.com; 5School of Physical Education and Sport Science, Division of Sport Medicine and Biology of Exercise, National and Kapodistrian University of Athens, 17237 Athens, Greece; ngeladas@phed.uoa.gr; 6Department of Nutrition and Dietetics, Harokopio University of Athens, 17671 Athens, Greece; lsidossis@kines.rutgers.edu; 7Department of Kinesiology and Health, Rutgers University, New Brunswick, NJ 08854, USA; 8Department of Technology, Sentio Solutions Inc., 11525 Athens, Greece; haris@myfeel.co; 9Department of Neurosciences and Precision Medicine, University Mental Health Research Institute “Costas Stefanis”, 11527 Athens, Greece; 102nd Neurological Department, Henry Dunant Hospital Center, 11526 Athens, Greece

**Keywords:** electrodermal activity (EDA), special forces, Navy SEALs, sympathetic response, stress resilience

## Abstract

Stress resilience plays a key role in task performance during emergencies, especially in occupations like military special forces, with a routine consisting of unexpected events. Nevertheless, reliable and applicable measurements of resilience in predicting task performance in stressful conditions are still researched. This study aimed to explore the stress response in the Hellenic Navy SEALs (HN-SEALs), using a cognitive–physiological approach. Eighteen candidates under intense preparation for their enlistment in the HN-SEALs and 16 healthy controls (HCs) underwent Stroop tests, along with mental-state and personality examination. Simultaneously, electrodermal activity (EDA) was assessed during each one of cognitive testing procedures. Compared to healthy control values, multiple components of EDA values were found decreased (*p* < 0.05) in the HN-SEALs group. These results were associated with an increase in resilience level in the HN-SEALs group, since a restricted sympathetic reactivity according to the reduced EDA values was observed during the stressful cognitive testing. This is the first report providing physiological measurements of the sympathetic response of HN-SEALs to a stressful situation and suggests that EDA turns out to be a simple and objective tool of sympathetic activation and it may be used as a complementary index of resilience in HN-SEALs candidates.

## 1. Introduction

Hellenic Navy SEALs (named for the environment they operate in: sea, air, and land) is an elite military force responsible for special activities, working in disorderly and unknown surroundings with narrow margin for error. Training to become a Hellenic Navy SEAL is known to be one of the most challenging training regimens in the special forces. The central component of this training involves the accomplishment of Basic Underwater Demolition/SEAL (BUD/S) First Phase. During this phase, candidates are tested in circumstances of extreme physical and mental stress for 7 weeks. The most exhausting part of BUD/S takes place in week 4, known as “Hell week”, in which candidates undergo extraordinary sleep deprivation (being given only 45 min of sleep per night) while completing demanding tasks and exercises throughout five and a half days of practicing [1]. 

Less than 20% of candidates who start BUD/S training successfully finish it. Candidates’ stress reactions may be particularly important in this extremely stressful situation. Instructors communicate messages about embracing stress in training (colloquially mentioned as “embrace the suck”) and at the same time spare no effort to intensify candidates’ stress throughout training to imitate combat surroundings. The aim behind the structure and the intensity of the BUD/S is to create training conditions similar to real combat, a situation that unavoidably releases high levels of stress. It includes unexpected attacks and uncontrolled threats that demand attention and rapid decision making. Under these circumstances, the sympathetic nervous system and the “fight or flight” response are activated [2,3,4,5]. Acute stress response leads to certain physiological and psychological reactions which prepare the individual to face combat’s hardships. Candidates who react to stress with readiness and show alertness as much as resilience may show greater perseverance and performance throughout training. Those resisting stressful attempts may feel powerless to manage the increasing pressure, so negative effects may also arise when this reaction is maintained in time [6].

Electrodermal activity (EDA) has long been considered as a reliable measurement of physiological and mental stress [7], since it is a validated marker of sympathetic activity [8,9]. EDA or galvanic skin response (GSR) measures the electrical resistance of the skin. EDA increases in response to a stressor, while skin resistance decreases. It is obvious that the activation of the skin glands as a result of the sympathetic nervous system response to a stressful stimulus is responsible for the augmented skin conductivity during a stress reaction; thus, EDA is an objective measurement of autonomic nervous system activation [9]. Furthermore, numerous studies in patients with discrete brain lesions and functional imaging methods have clarified the contribution of specific brain regions in EDA alterations [10,11,12]. According to these studies, various aspects of arousal linked to different brain regions (ventromedial prefrontal cortex, insula and amygdala) may mediate autonomic response and alter EDA [9]. Three physiological pathways are mainly involved in the production of EDA signals: pathways under hypothalamic control, pathways influenced by contralateral and unilateral basal ganglion (one of them connected with the premotor cortex and the other with the frontal cortex), and pathways guided by the reticular formation in the brainstem [13]. Every pathway implies a different functional role: The hypothalamic pathway controls sweating, amygdala activation indicates affective processes, premotor cortex activation is required in fine motor control, prefrontal cortex activation is essential for orienting and attention, and activation of the reticular formation is associated with increased muscle tone [13,14].

Psychophysiological responses of the military population, and especially of the special forces, in stressful conditions are inadequately researched, and the relevant literature is limited and controversial.

On 2007, Morgan and his colleagues concluded that reduced vagal tone was associated with enhanced performance in male military personnel exposed to high stress [15]. This result was attributed to better cognitive functioning and emotion regulation with suppressed vagal tone.

On the other hand, Clemente-Suárez and Robles-Pérez, after analyzing psychophysiological changes in a simulated urban combat [16] concluded that a high sympathetic nervous system stimulation is unavoidably produced in such a stressful situation. Other studies have compared elite and non-elite soldiers [3], acute high-stress combat situations in professional soldiers [4] and autonomic and cortical response of soldiers in different combat scenarios [6] with various results.

Furthermore, a recent interesting trend in research for general military and special forces seems to be the discovery of prognostic factors for successful performance in extraordinary stressful conditions. The recent article “Stress, Mindsets, and Success in Navy SEALs Special Warfare Training” by Smith et al. [1] concludes that specific mindsets, such as stress-is-enhancing mindsets—the belief that stress enhances health and performance—predict greater persistence through training. Psychological and physiological predictors of success in the US Army Special Forces [17] and in Navy SEALs [18] have been recently studied. Resilience score, grit, lower C-reactive protein, higher cortisol and sex-hormone binding globulin (SHBG) predicted better stress performance and resilience according to the first study, while, likewise, resilience score and high cortisol level, as well as DHEA (dehydroepiandrosterone) and brain-derived neurotrophic factor (BDNF) levels, were associated with stress adaptation in the second study.

The aim of the present study is to compare the performance on the neuropsychological tasks between the HN-SEALs and healthy controls (HCs) while measuring their EDA reactions. We hypothesize that HN-SEALs may show decreased sympathetic reaction in EDA measurements throughout the cognitive and psychological testing due to their enhanced stress resilience throughout their training. To our knowledge, this is the first study that measures EDA in HN-SEALs during cognitive testing and might contribute to the worldwide effort of stress adaptation predicting factors detection in special forces.

## 2. Materials and Methods

The study was conducted in accordance with the Declaration of Helsinki and was approved by the Ethics Committee of Harokopio University of Athens. It was finally conducted after review and approval by the Hellenic Navy General Staff.

Eighteen HN-SEALs candidates (age 24.2 ± 3.4 years, height 180.4 ± 1.7 cm, body mass index 24.4 ± 0.2) and 16 HCs with matching demographic and body characteristics participated in the study. The HN-SEALs participated in the Basic Underwater Demolition SEAL (BUD/S) training of the Hellenic Navy Special Operations Command from September 2018 to April 2019. This study was conducted right before the most demanding military training week (Hell Week) of the BUD/S.

The study was conducted at the Laboratory of Psychophysiology of the 1st Psychiatric department of the National and Kapodistrian University of Athens.

Each participant was entering the laboratory alone. Afterwards, a wristband was placed on their wrist for the psychophysiological measurement. Finally, they were given two questionnaires and took part in three cognitive tasks in a randomized order. The total duration of the procedure was 30 min.

The following psychometric and neuropsychological tools were used:

Symptoms Checklist 90 Revised (SCL-90R): It is used as a screening tool for current mental state. It is a self-report instrument. It consists of 90 questions divided in nine subcategories: somatization, obsessive compulsive, interpersonal vulnerability, depression, anxiety, hostility, phobic anxiety, paranoid ideation and psychoticism. All the subcategories together describe psychological, behavioral and physical symptoms. Its scoring is based on a 5-point Likert scale from 0 = not at all to 4 = very much. In addition to the score extracted for each subcategory, there are three more indices: the global severity index, the positive symptom distress index and the positive symptom total [19].

Eysenck Personality Questionnaire (EPQ): It is a three-dimensional personality-assessment tool. It consists of 84 questions: 24 of them define the dimension of psychoticism (P), 22 the dimension of neuroticism (N), 19 the extraversion (E) and 19 the dimension of lie (L) [20].

Color–Word Stroop: Color–Word Stroop is a neuropsychological test measuring the executive functions and more specifically the inhibition control. It was first presented by John Ridley Stroop on 1935. It has been proven that this task is very difficult to do and leads to slow error-prone responding due to the phenomenon of interference. In the most common version of the test, which is used in this study, subjects are instructed to read three different tables as fast as possible. Every table consists of 100 words. The first table (c1) includes color–words (printed names of colors) written in black ink and repeated in random order. The participants must read these color–words. The second table (c2) consists of patches of symbols printed in different colors, while the subjects must name the color patches. Conversely, in the third table (c3), color–words are printed in an inconsistent color ink (for instance the word “blue” is printed in red ink). In this incongruent condition, participants are required to name the color while ignoring the word itself. For every table, they have 1 min to read as many words or colors as possible. For every participant, we keep track of the number of words read, the number of mistakes made and the number of mistakes made and then corrected. It has been proven that this task leads to slow error-prone responding due to the phenomenon of interference. Actually, the participants are required to perform a less automated task (naming ink color) while inhibiting the interference arising from a more automated task (reading the word). This difficulty in inhibiting the more automated process is called the Stroop effect. Thus, the Color–Word Stroop is widely used to measure the ability to inhibit cognitive interference. Besides, its application is reported to measure other cognitive functions, such as attention, processing speed, cognitive flexibility and working memory [21].

Number Stroop: Number Stroop is based on the principles of Color–Word Stroop, but instead of using stimuli of words and colors, it measures the interference phenomenon by using size and value. The present study used a computerized version of the test. In total, three conditions were created: congruent condition (the bigger number in size was also the bigger in value), incongruent condition (the bigger number in size was the smaller in value) and the neutral condition (numbers had the same size in the size comparison test and the same value in the value comparison test). Each participant had to peak the bigger of two numbers presented on a computer screen, either according to its size or according to its value, by pressing the corresponding arrow on the keyboard. Each test (size or value) was presented in a random order and it consisted of 180 repetitions. These 180 repetitions were divided in 6 groups of 30 comparisons, with a fixation point being presented on the screen for 500 ms between every group. For every participant, we measured the reaction time and the number of mistakes made [22].

Emotion Stroop: Emotion Stroop measures how the magnitude of an emotional reaction in words believed to have a negative meaning interferes in the execution of a task irrelevant to these words. Studies have showed that reaction time in negative words is bigger in comparison to neutral words. This difference has been attributed to the effect that negativity has on humans. The present study used a computerized version of the task. During the procedure, a ring was divided into a number of colors. In the center of the ring, words were appearing, each of different color. Participants had to match the word’s color with the same color on the ring, by using the mouse. The task consisted of 60 negative and 60 neutral words; each word was presented randomly five times, each time painted in a different color [23].

Psychophysiological measurement assessed EDA in HN-SEALs and HCs as a sympathetic response index to the concomitantly accomplished neuropsychological evaluation.

In order to measure autonomic response, First Psychiatric Clinic of National and Kapodistrian University has developed a collaboration with Sentio Solutions Inc., which provides the Laboratory of Psychophysiology with a wristband called “Feel Emotion Sensor” (FES). Sentio’s proprietary technology makes it capable of measuring heart rate variability, electrodermal activity and temperature. Utilizing four built-in sensors in the wristband, FES collects information on the aforementioned biosignals and analyzes them, using the most modern technological methods. Advanced technologies related to artificial intelligence and signal-processing algorithms are in the position to track down emotional reactions while participants wear the wristband and execute various tasks.

EDA was expressed as mean skin conductance level in microsiemens (µS) between the different data points. The EDA data were acquired within the 30-min time interval, in which the total neuropsychological testing was accomplished.

Additionally, we present the basic features and metrics of an EDA signal that we assessed in our study.

The most notable feature of an EDA signal is the appearance of skin conductance responses (SCRs) secondary to an underlying sympathetic response to a stimulus. The SCRs are divided in rapid and in flat transient events distinguishable in the EDA signal. The combined total of SCRs makes up the phasic EDA component. Quantitative measures of the SCRs are used to assess a subject’s response to unexpected stimuli or stimulant (tonic) tests (qualitative workload or cognitive stress). SCRs amplitude and the onset-to-peak time are the most useful quantitative measures in EDA [13].

The skin response to a tonic stimulus is measured by the skin conductance level (SCL) and the nonspecific skin conductance responses (NSSCRs) [14]. SCL relates to the total conductance obtained from the tonic component of EDA, refers to the slow shifts of the EDA (measurements obtained during a non- stimulation rest period) and is measured with the same units as EDA (μS). The NSSCRs are the number of SCRs in a period of time in the presence of a sustained stimulus; they are also considered a tonic measure of spontaneous fluctuations in EDA. SCL and NSSCRs are considered as the tonic component of EDA.

SCRs, either specific to a stimulus or spontaneous, are characterized by a rise from the initial level to a peak, followed by a decline. This rise is the amplitude of the SCRs (conductance at the peak relative to the conductance at the onset) and can reach a minimum of 0.04 μS to several μS. The rise time (time from the onset of the SCR to the peak) ranges between 0,5 s and 5 s [13]. When caused by a stimulus, the onset of the SCR (latency) is typically between 1 and 5 s after the delivery of the stimulus [14]. 

Values are presented as absolute and relative frequencies for nominal variables concerning demographic data and as mean and standard deviation for the continuous variables regarding the scales investigated in this work. Normality assumption was examined via Shapiro–Wilks test for both groups, Control and Experimental [24]. Comparisons between the aforementioned groups as regards the EDA levels were performed by using parametric and non-parametric tests with selected significance level of 5%. Specifically, two independent samples *t*-test were conducted for testing the equality of mean values between HN-SEALs and HCs for normally distributed variables, and the Mann–Whitney test was conducted for testing the equality of median values between HN-SEALs and HCs for non-normally distributed variables [25,26]. Data analysis was performed by using statistical software of IBM SPSS (Version 23) [27].

## 3. Results

Eighteen HN-SEALs and 16 HCs were included in the study. The demographic characteristics of the two groups are presented in Table 1. The mean age of the subjects was 28.65 years, while the majority of them was unmarried.

Regarding the psychometric evaluation, HCs had a statistically significantly lower mean level for the category of somatization (0.51 ± 0.38), compared to HN-SEALs (1.34 ± 0.61) (t (15,16) = − 4.706, *p* < 0.01), with a large size effect Cohen’s d, equal to 0.87 (Table 2). Furthermore, HCs had a statistically significantly lower mean level at the category of anxiety (0.78 ± 0.71), compared to HN-SEALs (1.26 ± 0.56) (t (15,16) = − 2.184, *p* = 0.037 < 0.05) with a large size effect Cohen’s d equal to 0.75 (Table 2), and statistically significantly higher mean level at the trait of neuroticism (10.63 ± 6.09), compared to HN-SEALs (7.00 ± 3.41) (t (15,16) = 2.093, *p* = 0.047 < 0.05), with a size effect Cohen’s d equal to 0.70.

No statistically difference was observed in any of the Stroop tasks between HCs and HN-SEALs (Table 3 and Table 4).

On the contrary, statistically significant differences with the independent sample *t*-test were found on the physiological measurements recorded during the neuropsychological tasks. More specifically, regarding the duration of an EDA response either in the tonic or phasic component during the incongruent condition (third table) of the Color–Word Stroop test was statistically significantly higher in the group of HCs (tonic: 779.07 ± 513.30 and phasic: 878.73 ± 559.25) than the HN-SEALs group (tonic: 323.45 ± 137.80 and phasic: 302.27 ± 110.72) (*p* < 0.01) (Table 5). Likewise, the minimum EDA value during the value Number Stroop test was significantly higher in the group of HCs (−1.83 ± 0.67) than the HN-SEALs group (−2.69 ± 0.82) (*p* < 0.01) (Table 5).

Similarly, during all the three tables of Color Stroop Test HCs showed statistically significantly (*p* < 0.05) higher mean level of all the EDA measurements (Mean first difference, Mean second difference, EDA range, Min tonic EDA, Recurrence plot metric of tonic EDA and fourth order statistic of tonic EDA) analyzed with the Mann–Whitney U test compared with the same EDA measurements in HN-SEALs (Table 6).

Furthermore, during the size Number Stroop test, HCs showed also statistically significantly (*p* < 0.05) higher mean level of the EDA measurements (Mean EDA, Max EDA, Area below an EDA response, Duration of an EDA response (phasic EDA component), Area below an EDA response (phasic EDA component), Max tonic EDA) analyzed with the Mann–Whitney U test compared to the same measurements in HN-SEALs.

In addition, during the Value Number Stroop, it was also remarked that HCs also showed a statistically significantly (*p* < 0.05) higher mean level of the Area below an EDA response (phasic EDA component) analyzed with the Mann–Whitney U test compared to HN-SEALs.

## 4. Discussion

The aim of this study was to investigate if EDA could be considered as a possible tool in measuring stress resilience in HN-SEALs compared with HCs. For this purpose, we assessed parameters belonging both to tonic and phasic components of EDA while the participants were simultaneously cognitively and psychologically tested. We used color–word and value Stroop tests as cognitive tests and emotional Stroop test as psychological test. We applied these tests while concomitantly we were measuring the EDA reaction in two separate groups of participants: HN-SEALs and HCs. We hypothesized that HN-SEALs would not differ from healthy controls on their performance on the cognitive tasks, whereas they might show higher resilience and adaptability in both mentally and emotionally stressful events, probably due to their long and intense training in stressful conditions.

Our results showed that HN-SEALs performance on both cognitive and emotional Stroop tasks was equivalent to HCs. Conversely, they conveyed statistically significant lower levels of EDA in many different EDA components assessed during the simultaneous cognitive and emotional evaluations.

More specifically, regarding the duration of an EDA response during the incongruent condition (third table) of the Color–Word Stroop test was statistically significantly higher in the group of HCs than the HN-SEALs group (*p* < 0.01) (Table 5). Likewise, the minimum EDA value during the value Number Stroop test was significantly higher in the group of HCs than the HN-SEALs group (*p* < 0.01) (Table 5).

Additionally, during all the three tables of Color Stroop Test, HN-SEALs showed significantly lower mean level of all the EDA measurements compared with the HCs. Furthermore, during the size and value Number Stroop test, HN-SEALs also showed a significantly lower mean level of all the EDA measurements, as compared with the HCs.

In a task-related context, stress resilience is described as the capacity of keeping normal physical and psychological functioning, when put at risk of exceptional amounts of distress and trauma [3]. Stress resilience is critical in the context of a highly demanding and stressful profession such that of an HN-SEAL.

According to the research background provided in a comprehensive review [2] and in most of the recent literature [10,28,29] metrics based on the objectively measurable responses of the peripheral physiology, such as EDA, focus on the psychophysiological concepts mostly related to stress resilience, such as sympathetic functioning. Furthermore, physiological features like EDA are definitely low-cost and non-invasive, compared to high-cost high-technology metrics (fMRI, gene expressions, etc.) and more objective than traditional psychometric tools for stress resilience assessment, which are susceptible to self-report bias. Thus, psychophysiological tools are considered as the most valuable indicators of stress resilience research in special and extended populations.

Psychophysiological responses of the military population and especially of the special forces in stressful conditions are inadequately researched, and the relevant literature is limited and controversial. The current literature is expanding in this area, and stress resilience in special military populations seems to be becoming more intensively studied [2,10,17,18,29].

Psychological and physiological predictors of success in the US Army Special Forces [17] and in Navy SEALs [18] have been recently studied. Resilience score, grit, lower C-reactive protein, higher cortisol and sex-hormone binding globulin (SHBG) predicted better stress performance and resilience according to the first study, while, likewise, resilience score and high cortisol level, as well as DHEA (dehydroepiandrosterone) and brain-derived neurotrophic factor (BDNF) levels, were associated with stress adaptation in the second study.

Controversial conclusions were also recently published about the EDA measurements in highly demanding occupational conditions. EMG- and EDA-based traditional startle response measures, which are considered as potential indicators of stress resilience, have been shown to contribute to job-specific task performance under realistic occupational stress in a population of 40 air traffic control (ATC) candidates [29]. The authors suggest that the main practical aim of this kind of research that performs assessment of validated objective physiological features of stress resilience, as well as determining limits of the performance envelope in realistic occupational settings, is to complement the existing criteria for selection of ATC candidates and of other candidates for highly stressful occupations. On the contrary, in another recent study that used physiological measurements during a stressful task, increased arousal of the participants was remarked. Salivary a-amylase and EDA were investigated in 12 drone operators in service over a 2-h operating flight. Compared to baseline values, EDA and sAA values increased in operating conditions and were also associated with an increase in anxiety level. The authors suggest that this is likely related to increased arousal and perceived self-control, and might facilitate an increase in performance and self-confidence [30].

Our study supports the approach of the study by Sarlija et al., 2020 [29], since our results also indicate lower overall levels of EDA under stressful cognitive stress in HN-SEALs compared with HCs.

A basic limitation of the current research is that results have been obtained from a limited number of subjects. This prevents us from generalizing the conclusions. It is essential to support them through follow-up studies.

However, to our knowledge, our study is the first study that measures EDA in HN-SEALs during neuropsychological testing and might contribute to the worldwide effort of stress adaptation predicting factors detection in special forces.

## 5. Conclusions

Overall, we conclude that HN-SEALs show increased stress resilience according to EDA measurements during stressful cognitive assessments. We also suggest that EDA may be used in a complementary way with other physiological measurements and the existing selection criteria for candidates for the highly selective groups of special military forces and, generally, in occupations demanding effective functioning under dangerous and unexpected conditions, as an index of stress resilience in these groups.

## Figures and Tables

**Table 1 ijerph-18-04384-t001:** Demographic characteristics of participating subjects per group.

Demographic Characteristics		All	HN-SEALs	HCs
Number of participants		34 (100%)	18 (52.9%)	16 (47.1%)
Age (years)		28.65 ± 9.76	24.17 ± 3.35	33.69 ± 12.08
Years of Education		15.42 ± 1.75	15.76 ± 0.97	15.06 ± 2.29
Family Status	Unmarried	28 (82.4%)	16 (88.9%)	12 (75.0%)
	Married	5 (14.7%)	2 (11.1%)	3 (18.8%)
	Divorced	1 (2.9%)	0 (0.0%)	1 (6.3%)
Right-handed		33 (97.1%)	18 (100%)	15 (93.8%)
Health Problem (No)		34 (100%)	18 (100%)	16 (100%)

**Table 2 ijerph-18-04384-t002:** Independent sample *t*-test for equality of SCL and EPQ mean values between HN-SEALs and HCs.

Variable	HCs (N = 16) (Mean ± SD)	HN-SEALs (N = 18) (Mean ± SD)	Independent Samples’ *t*-Test	Effect Size Cohen’s d
SCL-90R				
Somatization	0.51 ± 0.38	1.34 ± 0.61	−4.706 **	0.87
Anxiety	0.78 ± 0.71	1.26 ± 0.56	−2.184 *	0.75
EPQ				
Neuroticism	10.63 ± 6.09	7.00 ± 3.41	2.093 *	0.70

* *p* < 0.05, ** *p* < 0.01.

**Table 3 ijerph-18-04384-t003:** Independent sample *t*-test for equality of Stroop mean values between HN-SEALs and HCs.

Variable	HCs (N = 16) (Mean ± SD)	HN-SEALs (N = 18) (Mean ± SD)	Independent Samples’ *t*-Test	Statistical Significance *p*-Value	Effect Size Cohen’s d
CWS sheet 1 words sum	135.13 ± 15.09	144.39 ± 18.06	−1.611	0.117	0.55
CWS sheet 2 words sum	94.38 ± 13.09	104.06 ± 14.93	−1.998	0.054	0.69
CWS sheet 3 words sum	62.00 ± 17.37	67.33 ± 14.94	−0.962	0.343	0.33
NS size RT congruent condition	0.91 ± 0.29	0.78 ± 0.08	1.806	0.081	0.64
NS size RT incongruent condition	0.99 ± 0.31	0.86 ± 0.10	1.554	0.139	0.59
NS size RT neutral condition	1.14 ± 0.37	0.94 ± 0.15	1.958	0.066	0.73
NS size mistakes congruent condition	2.40 ± 1.88	3.78 ± 2.53	−1.742	0.091	0.61
NS value RT congruent condition	0.77 ± 0.23	0.73 ± 0.11	0.807	0.426	0.23
NS value RT incongruent condition	0.83 ± 0.22	0.78 ± 0.12	0.924	0.362	0.29
NS value RT neutral condition	0.82 ± 0.22	0.76 ± 0.10	1.083	0.287	0.36

**Table 4 ijerph-18-04384-t004:** Mann–Whitney U test for equality of Stroop median values between HN-SEALs and HCs.

Variable	HCs (N = 16) (Median ± Interquartile Range)	HN-SEALs (N = 18) (Median ± Interquartile Range)	Mann–Whitney U	Statistical Significance *p*-Value	Effect Size Cohen’s d
CWS sheet 1 mistakes	0.00 ± 0.00	0.00 ± 0.00	130.50	0.646	0.16
CWS 1 sheet 1 fixed mistakes	0.00 ± 0.00	0.00 ± 1.00	113.50	0.297	0.37
CWS sheet 2 fixed mistakes	1.00 ± 2.00	1.00 ± 2.25	116.00	0.347	0.34
CWS sheet 3 mistakes	0.00 ± 0.00	0.00 ± 1.00	130.00	0.646	0.17
CWS sheet 3 fixed mistakes	0.00 ± 1.00	1.00 ± 3.00	102.50	0.154	0.51
NS value mistakes congruent condition	0.00 ± 0.00	0.00 ± 0.25	122.00	0.656	0.16
NS value mistakes incongruent condition	1.00 ± 3.00	1.00 ± 2.00	109.00	0.361	0.33
NS value mistakes neutral condition	0.00 ± 1.00	0.00 ± 1.00	129.00	0.845	0.08
ES total RT for negative words	0.97 ± 0.30	0.93 ± 0.13	110.00	0.561	0.22
ES mistakes for negative words	0.00 ± 1.00	0.00 ± 1.00	114.00	0.667	0.16
ES total RT for neutral words	0.96 ± 0.28	0.95 ± 0.15	118.00	0.750	0.12
ES mistakes for neutral words	0.00 ± 0.25	0.00 ± 0.25	122.00	0.896	0.07

**Table 5 ijerph-18-04384-t005:** Independent sample *t*-test of electrodermal activity mean values during cognitive tasks in HN-SEALs and HCs.

Variable	HN-SEALs (N = 18) (Mean ± SD)	HCs (N = 16) (Mean ± SD)	Independent Samples’ *t*-Test
Color–Word Stroop sheet 3			
*Duration of an EDA response*	323.45 ± 137.80	779.07 ± 513.30	−3.280 **
*Duration of an EDA response (phasic EDA component)*	302.27 ± 110.72	878.73 ± 559.25	−3.889 **
Number Stroop value			
*Minimum EDA value*	−2.69 ± 0.82	−1.83 ± 0.67	−2.778 *

* *p* < 0.05, ** *p* < 0.01.

**Table 6 ijerph-18-04384-t006:** Mann–Whitney U test for equality of electrodermal activity mean values during cognitive tasks between HN-SEALs and HCs.

Variable	HCs (N = 16) (Median ± Interquartile Range)	HN-SEALs (N = 18) (Median ± Interquartile Range)	Mann–Whitney U
Color–Word Stroop sheet 1			
*Mean 1st difference*	0.02 ± 0.01	0.01 ± 0.01	36.00 *
*Mean 2nd difference *	0.03 ± 0.02	0.02 ± 0.02	36.00 *
*EDA range*	4.23 ± 1.71	3.48 ± 0.73	44.00 *
*Min tonic EDA*	0.28 ± 0.57	0.09 ± 0.17	38.00 *
Color–Word Stroop sheet 2			
*Recurrence plot metric of tonic EDA*	53.89 ± 46.77	94.21 ± 76.44	44.50 *
Color–Word Stroop sheet 3			
*4th order statistic of tonic EDA*	2.11 ± 0.80	2.48 ± 1.32	36.00 *
*Recurrence plot metric of tonic EDA*	0.01 ± 0.02	0.00 ± 0.00	40.00 *
Number Stroop size			
*Mean EDA*	0.42 ± 0.73	0.11 ± 0.16	31.00 *
*Max EDA*	0.44 ± 0.79	0.16 ± 0.18	31.00 *
*Area below an EDA response*	207.49 ± 923.98	54.13 ± 93.05	31.00 *
*Duration of an EDA response (phasic EDA component)*	983.50 ± 658.00	269.50 ± 822.25	29.00 *
*Area below an EDA response (phasic EDA component)*	3.07 ± 18.13	0.59 ± 1.65	25.00 *
*Max tonic EDA*	0.43 ± 0.78	0.14 ± 0.17	31.00 *
Number Stroop value			
*Area below an EDA response (phasic EDA component)*	4.02 ± 14.40	0.59 ± 2.61	27.00 *

* *p* < 0.05.

## Data Availability

The datasets used and analyzed during the current study are available from the corresponding author, under reasonable request.

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
