# Peer review of "Assessment of Resilience of the Hellenic Navy Seals by Electrodermal Activity during Cognitive Tasks"

_ijerph, 2021, doi:10.3390/ijerph18084384_

Round 1

Reviewer 1 Report

I found this work well written and interesting.

I have very minor aspects to ask to the Authors.

The Authors use the acronymous EDA for Skin conductance. This acronym stands for Electrodermal Activity, which is another name for skin conductance or galvanic response. However, some readers might find it confounding. For this reason, I ask the Authors to change “skin conductance" from line 205 in “Electrodermal Activity”.

What do the Authors mean with “tonic” stimuli? Usually, responses can be “tonic” or “phasic”, and the usual terminology is that the skin conductance level is tonic, while the skin conductance response is the phasic answer.

Do the Authors have any reason to use this unusual (at least for me) terminology? If it is not the case, please change it so a reader will not be confounded.

Please, in your analysis also add the effect sizes.

Author Response

Reviewer 1:

Reviewer’s 1 suggestion:

The Authors use the acronymous EDA for Skin conductance. This acronym stands for Electrodermal Activity, which is another name for skin conductance or galvanic response. However, some readers might find it confounding. For this reason, I ask the Authors to change “skin conductance" from line 205 in “Electrodermal Activity”.

Writer’s response:

This expression is deleted in the final manuscript.

Reviewer’s 1 suggestion

What do the Authors mean with “tonic” stimuli? Usually, responses can be “tonic” or “phasic”, and the usual terminology is that the skin conductance level is tonic, while the skin conductance response is the phasic answer.

Do the Authors have any reason to use this unusual (at least for me) terminology? If it is not the case, please change it so a reader will not be confounded.

Please, in your analysis also add the effect sizes.

Writer’s response:

Following the reviewer’s suggestions we added a more comprehensive analysis of the tonic and phasic componenents of EDA and the effect sizes of our the main measurements of the method. The relevant information can be found in the new manuscript between lines 166-185.

We make an effort to generally improve the description of our introduction, methods, results and discussion sections.

Reviewer 2 Report

The authors measured electrodermal activity (EDA) in the Hellenic Navy SEALs (HN-SEALs) candidates and the healthy controls during stressful cognitive tests. The EDA characteristics showed statistically significant differences between the HN-SEALs candidates and the healthy controls. Based on this observation, they suggest that the EDA may be used as a complementary index of stress resilience. 

The aim of this study is interesting and important. However, I think that the physiological interpretation of the observed results in the discussion part is very weak. That is, it would be difficult to convince the association of the observed EDA changes with the stress resilience. Thus, more systematic analysis of the observed results would be required. For instance, the correlations between the cognitive test results and the EDA characteristics are not shown. I think that the number of EDA parameters used in this study is too many. Probably, most of EDA parameters are strongly correlated each other.  Moreover, heart rate, heart rate variability and temperature measured using the wrist-band device are not shown and not analyzed. If they want to assess the autonomic response, the heart rate variability parameters would be useful. 

Here are minor comments:
1. What do figure 3 in line 221 and figure 2 in line 230 on page 5 indicate? I was not able to find any figures in this paper.
2. It is difficult to understand the definition of the EDA variables, such as c1_EDA_f21. Illustrate the definition of the EDA variables, and explain the interpretation of those.
3. In this study, a statistical test was repeated multiple times. To avoid the multiple comparison problem, p-value correction would be necessary.

In conclusion I cannot recommend publication of this paper.

Author Response

REVIEWER’S COMMENT:

The aim of this study is interesting and important. However, I think that the physiological interpretation of the observed results in the discussion part is very weak. That is, it would be difficult to convince the association of the observed EDA changes with the stress resilience. Thus, more systematic analysis of the observed results would be required. For instance, the correlations between the cognitive test results and the EDA characteristics are not shown. I think that the number of EDA parameters used in this study is too many. Probably, most of EDA parameters are strongly correlated each other. 

AUTHOR’S RESPONSE:

We have rewritten the whole discussion part and the analyzing of the results trying to clarify the correlations between results and resilience in our rewritten manuscript according to your expectations.

REVIEWER’S COMMENT:

Moreover, heart rate, heart rate variability and temperature measured using the wrist-band device are not shown and not analyzed. If they want to assess the autonomic response, the heart rate variability parameters would be useful. 

AUTHOR’S RESPONSE

Although we definitely agree with you for the usefulness of heart rate variability in assessing autonomic response, in this particular research Heart rate variability is separately analyzed in another parallel study, concluding in the same results in an attempt to support the idea that each method may assess autonomously the sympathetic response.

MINOR REVIEWER’S COMMENT:
1. What do figure 3 in line 221 and figure 2 in line 230 on page 5 indicate? I was not able to find any figures in this paper.

AUTHOR’S RESPONSE

We have removed any figure and figure numbers.

  1. It is difficult to understand the definition of the EDA variables, such as c1_EDA_f21. Illustrate the definition of the EDA variables, and explain the interpretation of those.

AUTHOR’S RESPONSE

We have replaced these EDA variables with their interpretations in the tables.

  1. In this study, a statistical test was repeated multiple times. To avoid the multiple comparison problem, p-value correction would be necessary.

AUTHOR’S RESPONSE

We attempted to limit the multiple comparison problem.

On the whole, we attempted a general improvement in our manuscript trying to clarify the methods, the results and main conclusions.